# FAIRLY EXPLAINING MONOTONIC MODELS: A NEW SHAPLEY VALUE

## ABSTRACT

The Shapley value has been widely used as an attribution method for explaining black-box machine learning models. A rigorous mathematical framework based on a number of axioms has enabled Shapley value to disentangle the black-box structure of models. Recent studies have shown that domain knowledge is an important component of machine learning models. Science-informed machine learning models that incorporate domain knowledge have demonstrated better generalization and interpretation capabilities. But do we obtain consistent scientific explanations when we apply attribution methods to science-informed machine learning models? In this study, we show that Shapley value cannot be guaranteed to reflect domain knowledge, such as monotonicity. To remedy Shapley's monotonicity failure, we propose a new version of Shapley value. As a result of extensive analytical and empirical examples, we show that Shapley value often produces misleading explanations for monotonic models, which can be avoided using the new method.

## 1 INTRODUCTION

In recent decades, machine learning (ML) models have achieved great success. As a part of the effort to facilitate the use of ML, explanation methods are provided to assist people in disentangling the black-box nature of ML. This study examines attribution problems, which involve the interpretation of feature importance to prediction. There have been a number of successful works in this direction (Lundberg & Lee, 2017; Ribeiro et al., 2016; Horel & Giesecke, 2020; Sundararajan et al., 2017).

The Shapley value (Shap) is one of the most popular methods for solving attribution problems (Shapley et al., 1953). A major advantage of the Shap is that it provides a fair contribution of features within a rigorous theoretical framework by satisfying some desired axioms. A rigorous foundation has provided people with the confidence to implement Shap. However, despite extensive analysis of axioms, these studies have largely focused on axioms for general models (Sundararajan & Najmi, 2020; Lundstrom et al., 2022; Friedman & Moulin, 1999).

Science, on the other hand, has been developed over many centuries. Consequently, a variety of domain knowledge has been developed for various fields. A number of studies have demonstrated that physics-informed machine learning (Karniadakis et al., 2021; Greydanus et al., 2019) improved black-box ML models in terms of interpretation and accuracy by enforcing conservation laws, for example. Finance and other applications often require monotonicity. A person's credit score should be decreased when there is one more past due balance on the account, for example. It is possible to achieve better generalization and interpretation when monotonicity is successfully enforced (Liu et al., 2020; Milani Fard et al., 2016; You et al., 2017; Repetto, 2022; Runje & Shankaranarayana, 2023). These models can be categorized as science-informed machine learning models.

In this paper, we ask the following question: **Can attribution methods deliver consistent scientific explanations if models contain certain scientific knowledge? If so, to what extent?** We focus on monotonicity as a common domain knowledge in practice. There are two types of monotonicity (Chen & Ye, 2023; Gupta et al., 2020). Besides commonly known individual monotonicity, **pairwise monotonicity** specifies that certain characteristics are intrinsically more important than others. As an example, in credit scoring, the number of past dues more than two months should be more significant than the number of past dues between one and two months. For related applications, monotonicity is usually a hard requirement, since it is closely related to fairness. As an example, a

fair credit scoring system should punish each additional late payment. Unfortunately, when it comes to the explanation for monotonic models, we find that Shap fails to reflect pairwise monotonicity.

This paper analyses monotonicity in greater detail and proposes a new version to remedy Shap's failure, namely the generalized monotonic Shapley value (GMShap). In recognition of the lack of classical Shap, we modify the game setting and propose additional axioms when pairwise monotonicity is involved. Accordingly, GMShap is uniquely determined under certain assumptions, in the same way as Shap. As a result of extensive analytical and empirical examples, we demonstrate that, when pairwise monotonicity is involved, Shap can often produce misleading explanations and produce unfair interpretations. Fortunately, GMShap has avoided these issues and has been able to provide reasonable and reliable explanations.

**Related Work.** There has been extensive discussion of axioms for attribution methods ( Lundstrom et al., 2022; Sundararajan et al., 2017; Sundararajan & Najmi, 2020; Friedman & Moulin, 1999; Xu et al., 2020). However, these studies mainly focused on the axioms of general models without domain knowledge. As for domain knowledge, individual monotonicity is considered in Sundararajan & Najmi (2020); Friedman & Moulin (1999), but no consideration is made of pairwise monotonicity. To the best of our knowledge, our work is the first analysis of general monotonic models.

On the other direction, Shapley values with a coalition structure have also been considered in the past Kamijo (2009); Grabisch & Roubens (1999); Owen (1977). These studies, however, also focus on somewhat general assumptions about coalition structure, whereas we consider coalition structures that are characterized by strong pairwise monotonicity.

## 2 PRELIMINARIES

### 2.1 ATTRIBUTION

For problem setup, assume we have $n$ features. For $\mathbf{a}, \mathbf{b} \in \mathbb{R}^n$, define $[\mathbf{a}, \mathbf{b}]$ to be the hyperrectangle. We denote a class of functions $f : [\mathbf{a}, \mathbf{b}] \to \mathbb{R}$ by $\mathcal{F}(\mathbf{a}, \mathbf{b})$, or simply $\mathcal{F}$. We assume $\mathbf{x} \in [\mathbf{a}, \mathbf{b}]$. Following Lundstrom et al. (2022), we call the point of interest $\mathbf{x}$ to explain as an explicand and $\mathbf{x}'$ a baseline. For simplicity, we assume $\mathbf{x} \geq \mathbf{x}'$, i.e., $x_i \geq x_i', \forall i$. We assume $\mathbf{x}' = \mathbf{0}$ unless otherwise stated. The Baseline Attribution Method is defined here.

**Definition 2.1** (Baseline Attribution Method (BAM)). *Given* $\mathbf{x}, \mathbf{x}' \in [\mathbf{a}, \mathbf{b}]$, $f \in \mathcal{F}(\mathbf{a}, \mathbf{b})$, *a baseline attribution method is any function of the form* $\mathbf{A} : [\mathbf{a}, \mathbf{b}] \times [\mathbf{a}, \mathbf{b}] \times \mathcal{F}(\mathbf{a}, \mathbf{b}) \to \mathbb{R}^n$. *We may also write* $\mathbf{A}$ *and denote* $A_i$ *as the $i$th attribution of* $\mathbf{A}$ *for simplicity.*

We review classical Shapley values and Integrated Gradients. Both can be considered to be members of the Shapley value family (Sundararajan & Najmi, 2020).

#### 2.1.1 (BASELINE) SHAPLEY VALUE

The Shapley value (Shap), introduced by Shapley et al. (1953), concerns the cooperative game in the coalitional form $(N, v)$, where $N$ is a set of $n$ players and $v : 2^N \to \mathbb{R}$ with $v(\emptyset) = 0$ is the characteristic function. In the game, the marginal contribution of the player $i$ to any coalition $S$ with $i \notin S$ is considered as $v(S \cup i) - v(S)$. By considering a variety of axioms, the attribution of a player $i$ by Shap is given by:

$$s_i = \sum_{S \subseteq N \setminus i} \frac{|S|!(|N| - |S| - 1)!}{|N|!} (v(S \cup i) - v(S)). \tag{1}$$

Here, we focus on the Baseline Shapley value (BShap), analyzed in Sundararajan & Najmi (2020), which calculates

$$v(S) = f(\mathbf{x}_S; \mathbf{x}'_{N \setminus S}). \tag{2}$$

That is, baseline values replace the feature's absence. We denote BShap attribution by $\mathrm{BS}_i(\mathbf{x}, \mathbf{x}', f)$ and $\mathrm{BS}_i$ sometimes. Two reasons motivate us to focus on the BShap. First, as pointed out by Sundararajan & Najmi (2020), BShap is capable of preserving many desired axioms in contrast to SHapley Additive Explanations (SHAP) (Lundberg & Lee, 2017); second, BShap's setup is naturally applicable to our applications.

### 2.1.2 INTEGRATED GRADIENTS

Integrated Gradients, introduced by Sundararajan et al. (2017), is given below.

**Definition 2.2** (Integrated Gradients (IG))**.** *Given* $\mathbf{x}, \mathbf{x}' \in [\mathbf{a}, \mathbf{b}]$ *and* $f \in \mathcal{F}(\mathbf{a}, \mathbf{b})$*, the integrated gradients attribution of the* $i$*-th component of* $\mathbf{x}$ *is defined as*

$$IG_i(\mathbf{x}, \mathbf{x}', f) = (x_i - x_i') \int_0^1 \frac{\partial f}{\partial x_i} \left( \mathbf{x}' + t(\mathbf{x} - \mathbf{x}') \right) \, dt. \tag{3}$$

For simplicity, we often use IG$_i$ for IG$_i(\mathbf{x}, \mathbf{x}', f)$.

### 2.2 INDIVIDUAL AND PAIRWISE MONOTONICITY

Without loss of generality, we assume that all monotonic features are monotonically increasing throughout the paper. Suppose $\boldsymbol{\alpha}$ is the set of all individual monotonic features and $\neg\boldsymbol{\alpha}$ its complement, then the input $\mathbf{x}$ can be partitioned into $\mathbf{x} = (\mathbf{x}_{\boldsymbol{\alpha}}, \mathbf{x}_{\neg\boldsymbol{\alpha}})$. Individual monotonicity is defined.

**Definition 2.3** (Individual Monotonicity)**.** *We say* $f$ *is individually monotonic with respect to* $\mathbf{x}_{\boldsymbol{\alpha}}$ *if*

$$f(\mathbf{x}_{\boldsymbol{\alpha}}, \mathbf{x}_{\neg\boldsymbol{\alpha}}) \leq f(\mathbf{x}_{\boldsymbol{\alpha}}^*, \mathbf{x}_{\neg\boldsymbol{\alpha}}), \forall \mathbf{x}_{\boldsymbol{\alpha}}, \mathbf{x}_{\boldsymbol{\alpha}}^* \text{ s.t. } \mathbf{x}_{\boldsymbol{\alpha}} \leq \mathbf{x}_{\boldsymbol{\alpha}}^*, \forall \mathbf{x}_{\neg\boldsymbol{\alpha}}, \tag{4}$$

*where* $\mathbf{x}_{\boldsymbol{\alpha}} \leq \mathbf{x}_{\boldsymbol{\alpha}}^*$ *denotes the inequality for all entries, i.e.,* $x_{\alpha_i} \leq x_{\alpha_i}^*, \forall i$.

In practice, certain features are intrinsically more important than others. Analog to equation 4, we partition $\mathbf{x} = (x_\beta, x_\gamma, \mathbf{x}_\neg)$. Without sacrificing generality, we assume that $x_\beta$ has greater significance than $x_\gamma$. Lastly, we require that all features exhibiting pairwise monotonicity also exhibit individual monotonicity. Pairwise monotonicity can be categorized into two types: strong and weak. As a more general definition, weak pairwise monotonicity is presented below.

**Definition 2.4** (Weak Pairwise Monotonicity)**.** *We say* $f$ *is weakly monotonic with respect to* $x_\beta$ *over* $x_\gamma$ *if*

$$f(x_\beta, x_\gamma + c, \mathbf{x}_\neg) \leq f(x_\beta + c, x_\gamma, \mathbf{x}_\neg), \forall \mathbf{x}, \mathbf{x}^* \in [\mathbf{a}, \mathbf{b}] \text{ s.t. } x_\beta = x_\gamma, c > 0. \tag{5}$$

Weak pairwise monotonicity compares the significance of $x_\beta$ and $x_\gamma$ at the same magnitude. Example A.3 is provided in Appendix A.1. In addition, there is a stronger condition of pairwise monotonicity, known as strong pairwise monotonicity, which is independent of the condition that $x_\beta = x_\gamma$. Here is the definition.

**Definition 2.5** (Strong Pairwise Monotonicity)**.** *We say* $f$ *is strongly monotonic with respect to* $x_\beta$ *over* $x_\gamma$ *if*

$$f(x_\beta, x_\gamma + c, \mathbf{x}_\neg) \leq f(x_\beta + c, x_\gamma, \mathbf{x}_\neg), \forall x_\beta, x_\gamma, \forall \mathbf{x}_\neg, \forall c \in \mathbb{R}^+. \tag{6}$$

**Example 2.6.** *In credit scoring, consider* $x_1$ *and* $x_2$ *to count the number of past due payments more than two months and between one and two months. Then the probability of default is strongly monotonic with respect to* $x_1$ *over* $x_2$.

### 2.3 AXIOMS

Many desirable characteristics of an attribution technique have been identified in the literature. Interested readers are referred to Lundstrom et al. (2022); Sundararajan & Najmi (2020); Sundararajan et al. (2017) for detailed discussion. Here, we list axioms that are considered in this paper.

- Implementation Invariance: $\mathbf{A}$ is independent of the type of model implemented, but only from the mathematical mapping of the domain to the range of a true model. The definition here differs slightly from the one in Sundararajan et al. (2017). Our main difference lies in the fact that we emphasize the true model with potential discrete features, whereas the other definition emphasizes neural networks, for which the domain is continuous.

- Completeness: $\forall f \in \mathcal{F}, \mathbf{x}, \mathbf{x}' \in [\mathbf{a}, \mathbf{b}]$, we have

$$\sum_{i=1}^n A_i(\mathbf{x}, \mathbf{x}', f) = f(\mathbf{x}) - f(\mathbf{x}'). \tag{7}$$

- Linearity: For $\alpha, \beta \in \mathbb{R}$ with two functions $f, g \in \mathcal{F}$, we have

$$A_i(\mathbf{x}, \mathbf{x}', \alpha f + \beta g) = \alpha A_i(\mathbf{x}, \mathbf{x}', f) + \beta A_i(\mathbf{x}, \mathbf{x}', g). \tag{8}$$

- Dummy(a): We say a player is a dummy player if his/her marginal contribution to any coalition is zero. If player $i$ is a dummy player, then

$$A_i(\mathbf{x}, \mathbf{x}', f) = 0. \tag{9}$$

- Symmetry(a): We say that players $i, j \in N$ are symmetric in game $(N, v)$ if they make the same marginal contribution to any coalition. If players are symmetric, then

$$A_i(\mathbf{x}, \mathbf{x}', f) = A_j(\mathbf{x}, \mathbf{x}', f). \tag{10}$$

- Demand Individual Monotonicity (DIM): Suppose $f$ is individually monotonic with respect to $x_\alpha$. We say a BAM preserves demand individual monotonicity if for $\mathbf{x}^* = \mathbf{x} + c\mathbf{e}_i$, where $\mathbf{e}_i$ is 1 at $i^{th}$ entry and 0 elsewhere, we have

$$A_\alpha(\mathbf{x}^*, \mathbf{x}', f) \geq A_\alpha(\mathbf{x}, \mathbf{x}', f), \forall c \in \mathbb{R}^+. \tag{11}$$

## 3 MONOTONIC AXIOMS AND PRESERVATION

### 3.1 NEW MONOTONIC AXIOMS

Motivated by the types of monotonicity in Section 2.2, we would like to study axioms related to monotonicity in greater detail. In addition to DIM, three new monotonic axioms are proposed here.

**Definition 3.1** (**Average Individual Monotonicity (AIM)**). *Suppose $f$ is individually monotonic with respect to $x_\alpha$, then we say a BAM preserves average individual monotonicity if*

$$A_\alpha(\mathbf{x}, \mathbf{x}', f) \geq 0. \tag{12}$$

**Definition 3.2** (**Average Weak Pairwise Monotonicity (AWPM)**). *Suppose $f$ is weakly monotonic with respect to $x_\beta$ over $x_\gamma$, $x_\beta > x'_\beta$ and $x_\gamma > x'_\gamma$. Suppose for an explicand $\mathbf{x}$, we have $x_\beta = x_\gamma$. Then we say a BAM preserves weak pairwise monotonicity if*

$$\frac{1}{x_\beta - x'_\beta} A_\beta(\mathbf{x}, \mathbf{x}', f) \geq \frac{1}{x_\gamma - x'_\gamma} A_\gamma(\mathbf{x}, \mathbf{x}', f). \tag{13}$$

**Definition 3.3** (**Average Strong Pairwise Monotonicity (ASPM)**). *Suppose $f$ is strongly monotonic with respect to $x_\beta$ over $x_\gamma$, $x_\beta > x'_\beta$, and $x_\gamma > x'_\gamma$. Then we say a BAM preserves average strong pairwise monotonicity if*

$$\frac{1}{x_\beta - x'_\beta} A_\beta(\mathbf{x}, \mathbf{x}', f) \geq \frac{1}{x_\gamma - x'_\gamma} A_\gamma(\mathbf{x}, \mathbf{x}', f). \tag{14}$$

### 3.2 PRESERVATION AND FAILURE OF AXIOMS

We present preservation results by IG and BShap, whereas proofs are left in Appendix A.1.

**Theorem 3.4.** *IG preserves AIM, AWPM for $x'_\beta = x'_\gamma$, and ASPM, but doesn't preserve DIM.*

**Theorem 3.5.** *BShap preserves AIM, DIM, and AWPM for $x'_\beta = x'_\gamma$, but doesn't preserve ASPM.*

DIM is not preserved by IG, which can be considered a weakness. Example A.4 is provided in Appendix A.1. Fortunately, IG preserves AIM, which can be viewed as a weaker condition for maintaining individual monotonicity.

**Theorem 3.6.** *If a BAM preserves DIM, then it preserves AIM.*

Additionally, IG requires continuous and differentiable functions. In practice, however, discrete features are common. It is possible for models such as neural networks to work if discrete features are treated as continuous features. Nevertheless, this could violate the implementation invariance axiom when IG is applied. Example A.6 is provided in Appendix A.1.

A major weakness of BShap is that it does not preserve ASPM. In the following example, we compare BShap and IG. **A striking result is revealed by the example: BShap does not satisfy ASPM even for logistic regressions!**

**Example 3.7.** *Consider a two-dimensional logistic regression*

$$y = f(x_1, x_2) = \sigma(\alpha + \beta_1 x_1 + \beta_2 x_2),$$

*where $\sigma(z) = \frac{e^z}{1+e^z}$ and $\beta_1 \geq \beta_2$. Clearly, $y$ is strongly monotonic with respect to $x_1$ over $x_2$.*

*By IG, we calculate that*

$$\mathbf{IG} = \begin{bmatrix} \beta_1 x_1 \\ \beta_2 x_2 \end{bmatrix} \int_0^1 f(tx_1, tx_2)(1 - f(tx_1, tx_2))\, dt. \tag{15}$$

*By the result, not only ASPM is preserved, but the ratio between $x_1$ and $x_2$ is perfectly recognized.*

*For BShap, we have*

$$BS_1 - BS_2 = \sigma(\alpha + \beta_1 x_1) - \sigma(\alpha + \beta_2 x_2). \tag{16}$$

*As a result, whenever $x_1 \geq x_2$, $BS_1 \geq BS_2$, which is consistent with our expectation. However,*

$$\frac{BS_1}{x_1} - \frac{BS_2}{x_2} = \frac{x_2 - x_1}{2x_1 x_2}(\sigma(\alpha + \beta_1 x_1 + \beta_2 x_2) - \sigma(\alpha))$$
$$+ \frac{x_1 + x_2}{2x_1 x_2}(\sigma(\alpha + \beta_1 x_1) - \sigma(\alpha + \beta_2 x_2)). \tag{17}$$

*Note that if $x_1 > x_2$, then ASPM might be violated by BShap! For example, for $\alpha = -10$, $\beta_1 = 2$, $\beta_2 = 1$, and $\mathbf{x} = (3, 1)$, then for BShap, we have $\mathbf{BS} \approx \begin{bmatrix} 0.033 \\ 0.015 \end{bmatrix}$.*

## 4 STRONG MONOTONIC GAMES

We would like to suggest a new Shapley value that preserves all of the axioms described above. In particular, we would like to propose a new version of BShap that preserves ASPM. Our focus is on BShap since IG is naturally applied to continuous features. We begin by considering only features with strong pairwise monotonicity. Consider $f(\mathbf{x})$ with $f(\mathbf{x}') = 0$ where $\mathbf{x}' = \mathbf{0}$, $\mathbf{x} = (x_1, \ldots, x_m)$, $f$ is individual monotonic with all $x_i$, and $f$ is strongly monotonic with respect to $x_i$ over $x_{i+1}$, $i = 1, \ldots, m-1$. We further assume that $x_i \in \mathbb{R}^+$, $\forall i$. Cost-sharing problems commonly assume similar assumptions (see for example, Friedman & Moulin, 1999), and we find that it is a suitable assumption for our application.

### 4.1 MOTIVATION

We argue that Shap fails due to the limitation of characteristic functions $v$. Shap considers the marginal contribution of player $i$ to any coalition with $i \notin S$ as $v(S \cup i) - v(S)$. In the scenario of strong pairwise monotonicity, this definition of marginal contribution might not make sense. In Example 2.6, suppose we are interested in the explanation at $\mathbf{x} = (1, 1)$ for $x_1$, BShap considers the marginal contributions $f(1, 0) - f(0, 0)$ and $f(1, 1) - f(0, 1)$. This makes sense when $x_1$ is independent of $x_2$. However, in this case, it is more appropriate to consider marginal contributions resulting from the difference between one and two months of delay. In particular, we believe that $f(0, 2) - f(0, 0)$ is a more appropriate measure of the baseline contribution for $x_2$ and $f(1, 1) - f(0, 2)$ for the marginal contribution of $x_1$. Then, we could evenly split contributions based on magnitudes of $x_i$. In other words, we could calculate

$$\phi = \begin{bmatrix} \frac{1}{2}(f(0, 2) - f(0, 0)) + f(1, 1) - f(0, 2) \\ \frac{1}{2}(f(0, 2) - f(0, 0)) \end{bmatrix}.$$

### 4.2 MONOTONIC SHAPLEY VALUE

Motivated by the above argument, we propose a monotonic version of Shapley values. Suppose we have a game with $(\mathbf{x}, f, \mathbf{w})$, where $\mathbf{w} : \mathbb{R}^m \to \mathbb{R}^{m+1}$. As opposed to $v$, magnitudes of $x_i$ are important in our calculation and $\mathbf{w}$ calculates the following values.

$$w_i(\mathbf{x}, f) = \begin{cases} f\left(0, \ldots, 0, \sum_{j=1}^i x_j, x_{i+1}, \ldots, x_m\right), & \text{if } 1 \leq i \leq m, \\ 0, & \text{if } i = m+1. \end{cases} \tag{18}$$

Next, we provide the formula for the monotonic Shapley value.

**Definition 4.1** (**Monotonic Shap (MShap)**). *For the game* $(\mathbf{x}, f, \mathbf{w})$, *the attribution* $\phi_i$ *by Monotonic Shapley value is calculated by*

$$\phi_i(\mathbf{x}) = \begin{cases} 0, & \text{if } \sum_{j=1}^i x_i = 0, \\ x_i \sum_{j=i}^m \frac{w_j(\mathbf{x}) - w_{j+1}(\mathbf{x})}{\sum_{k=1}^j x_k}, & \text{otherwise.} \end{cases} \tag{19}$$

Next, we discuss the preservation of axioms by MShap and we leave proofs in Appendix A.2

**Lemma 4.2.** *MShap satisfies implementation invariance, linearity, completeness, average individual monotonicity, and average strong pairwise monotonicity.*

In Lemma 4.2, we can see that MShap preserves most of the proposed axioms. There are, however, three axioms that require special attention. We begin by discussing the dummy and symmetry axioms. As we measure marginal contributions differently, we require different axioms. The key difference here is that we consider the impacts of $\mathbf{x}$ and $f$ separately, whereas they are considered together in Shap.

**Definition 4.3** (Dummy(b)). *If* $\forall f \in \mathcal{F}$, $f(\mathbf{x}) = f(\mathbf{x}^*)$, *where* $\mathbf{x}_j^* = \mathbf{x}_j$ *except for* $i$ *for all* $\mathbf{x}, \mathbf{x}^*$, *then* $A_i(\mathbf{x}, \mathbf{x}', f) = 0$. *Furthermore, if* $x_i = x_i'$, *let* $g(x_1, \ldots, x_{i-1}, x_{i+1}, \ldots, x_m) = f(x_1, \ldots, x_m)$ *and* $\mathbf{h}(x_1, \ldots, x_m) = (x_1, \ldots, x_{i-1}, x_{i+1}, \ldots, x_m)$, *then* $A_i = 0$ *and for* $j \neq i$, $A_j(\mathbf{x}, \mathbf{x}', f) = A_j(\mathbf{h}(\mathbf{x}), \mathbf{h}(\mathbf{x}'), g)$.

**Definition 4.4** (Symmetry(b)). *We say* $f$ *is symmetric about* $x_k$ *and* $x_l$ *if for any* $k < l$, $f(\mathbf{x}) = f(\mathbf{x}^*)$ *where* $x_i = x_i^*$ *for* $i \neq j, k$ *and* $x_k + x_l = x_k^* + x_l^*$. *We say a BAM preserves symmetry(b) for* $x_k, x_k^* > x_k'$ *and* $x_l, x_l^* > x_l'$ *if*

$$\frac{1}{x_k - x_k'} A_k(\mathbf{x}, \mathbf{x}', f) = \frac{1}{x_l - x_l'} A_l(\mathbf{x}, \mathbf{x}', f). \tag{20}$$

**Lemma 4.5.** *MShap preserves dummy(b) and symmetry(b).*

The third case involves the demand individual monotonicity axiom. As discussed in Friedman & Moulin (1999), DIM is desired for some features, but not necessarily all features. Here is the MShap result for DIM.

**Lemma 4.6.** *MShap preserves demand individual monotonicity for* $x_m$.

We would like to interpret this result. For strong pairwise monotonic features, this may not be necessary, as demonstrated in Example A.7 provided in Appendix A.1. In this regard, it is also not observed in general. $x_m$, however, represents the baseline contribution among all features. Due to this, its contribution is somewhat indicative of the magnitudes of the total features by formulas. Therefore, demand individual monotonicity makes sense.

Last, we present the uniqueness result for MShap.

**Theorem 4.7.** *MShap is a unique mapping that satisfies dummy(b), completeness, linearity, average strong pairwise monotonicity, and symmetry(b) for strong monotonic games.*

**Example 4.8.** *We calculate the MShap following Example 3.7. By calculation, we have*

$$\frac{MS_1}{x_1} - \frac{MS_2}{x_2} = \frac{\sigma(\alpha + \beta_1 x_1 + \beta_2 x_2) - \sigma(\alpha + \beta_2 x_1 + \beta_2 x_2)}{x_1}, \tag{21}$$

*whereas the ASPM is preserved.*

## 5 A TWO-STEP GENERALIZED MONOTONIC SHAPLEY VALUE

To this end, we generalize the game with general features. We split features into ones with strong pairwise monotonicity and others $\mathbf{x} = (\mathbf{x}_P, \mathbf{x}_\neg)$. We don't have any restrictions on $\mathbf{x}_\neg$, but fixing any $\mathbf{x}_\neg$ with $g(\mathbf{x}_P) = f(\mathbf{x}_P, \mathbf{x}_\neg) - f(\mathbf{x}_P', \mathbf{x}_\neg)$, we require that $(\mathbf{x}_P, g, \mathbf{w})$ are strong monotonic games, therefore satisfying all assumptions in Section 4. Such a structure is sufficient for most applications, and more complex structures can be generalized if necessary.

## 5.1 FIRST STEP CALCULATION

We treat $\mathbf{x}_P$ as a single feature since they usually describe the same feature and this is also why these features are able to be compared directly. As in Example 2.6, both $x_1$ and $x_2$ describe the number of past dues. Therefore, we treat them in a similar manner to the Shap. We consider the game $(N, v)$ in coalitional form, where $v : 2^N \to \mathbb{R}$. It is important to note that $N$ differs from the classical Shapley values. In the case where there are $m$ monotonic features and $n$ overall features, then $N = \{\{1, \ldots, m\}, m+1, \ldots, n\}$. By allowing a player $i = \{1, \ldots, m\}$, dummy(a) and symmetry(a) can be generalized. Example A.15 of generalized dummy and symmetry can be found in Appendix A.3. Next, we calculate attributions $\Phi_i$ according to the classical Shap method with the exception that attributions $\Phi_{P,j}$ for features $j$ in $P$ are undetermined. We call it the generalized Shapley value (GShap), which has the uniqueness result the same as Shap. GShap directly determines features without strong pairwise monotonicity. Then we discuss strong pairwise monotonic features.

## 5.2 SECOND STEP CALCULATION

Now we wish to determine $\Phi_{P,j}$ for $j \in P$. We rewrite equation 1 for $i = P$ as

$$\Phi_{P,j} = \sum_{S \subseteq N \setminus P} \frac{|S|!(|N| - |S| - 1)!}{|N|!} \varphi_j(S). \tag{22}$$

Then, we need to determine $\varphi_j(S)$ with $\sum_j \varphi_j(S) = v(S \cup P) - v(S)$. It can be recognized as an attribution problem for strong monotonic games discussed in Section 4. Specifically, for each $S$, we focus on

$$g_S(\mathbf{x}_P) = f(\mathbf{x}_P, \mathbf{x}_S, \mathbf{x}'_{N \setminus P \cup S}) - f(\mathbf{x}'_P, \mathbf{x}_S, \mathbf{x}'_{N \setminus P \cup S}). \tag{23}$$

We propose the following axiom since there is a natural correspondence between the original game and its subgames.

**Definition 5.1** (Consistency Axiom). *We say the GShap for the game $(N, v)$ is consistent with subgames if the attribution of the game is calculated in the form of equation 22, where $\varphi_j(S)$ is the attribution of the subgame.*

Axioms must be satisfied for each subgame. Therefore, based on Theorem 4.7, we apply MShap to each subgame. As a result, we have the following formula.

**Definition 5.2** (**Generalized Monotonic Shapley Values (GMShap)**). *The generalized monotonic Shapley value (GMShap) calculates the attribution as*

$$\Phi_{i,j} = \begin{cases} \sum_{S \subseteq N \setminus i} \frac{|S|!(|N| - |S| - 1)!}{|N|!} (v(S \cup i) - v(S)), & \text{if } i \neq P, j = 1, \\ \sum_{S \subseteq N \setminus P} \frac{|S|!(|N| - |S| - 1)!}{|N|!} \phi_j(g_S), & \text{if } i = P, \end{cases} \tag{24}$$

*where $\phi$ is calculated based on Definition 4.1 and $g_S$ is defined in equation 23.*

Example A.16 of GMShap is given in Appendix A.3. It is straightforward to determine the uniqueness result as follows.

**Theorem 5.3.** *Given GShap, GMShap is a unique mapping that preserves consistency, dummy(b), completeness, linearity, symmetry(b), and average strong pairwise monotonicity for each subgame for strong pairwise monotonic features.*

## 6 EMPIRICAL EXAMPLES

We present three examples to demonstrate the use of GMShap with a comparison to Shap. In all experiments, the monotonic groves of neural additive models (MGNAMs) proposed in Chen & Ye (2023) are used, in which strong pairwise monotonicity is maintained. A detailed description of the data and models can be found in Appendix A.4. In the examples, we compare both attributions $\phi_i$ and average attributions $\frac{\phi_i}{x_i}$ for strong pairwise monotonic features $x_i > 0$.

## 6.1 CREDIT SCORING - GIVE ME SOME CREDITS

We use the Kaggle credit score dataset [1]. In this dataset, we focus on three delinquency features that quantify the number of past dues and their duration: 90+ days, 60-89 days, and 30-59 days. Without loss of generality, we denote them as $x_1$, $x_2$, and $x_3$. Based on domain knowledge, the probability of default should be strongly monotonic with respect to $x_1$ over $x_2$ and $x_2$ over $x_3$.

We consider the following explicand as an example of illustration:

$$\mathbf{x} = \begin{bmatrix} 5 & 2 & 2 & 4 & 4 & 11 & 1.01 & 0 & 30 & 0.57 \end{bmatrix}.$$

Attributions by Shap and GMShap are provided in Figure 1. Results for Shap and GMShap are somewhat similar, which is not surprising given that nonmonotonic features are calculated similarly. Below is a brief summary of the results. The two most important features are $x_1$ and $x_7$. It is clear that for $x_1$, five times past due with a 90-day delay indicates that the applicant has difficulty repaying; $x_7$ implies that the applicant uses his/her money over the credit limits to pay off debts and costs. $x_2$ and $x_9$ are the next two features that contribute to this calculation. In the case of $x_2$, two further 60-89 days past dues further increase its risk, and $x_9$ which is the age, indicating a large amount of past due is abnormal for a 30-year-old young person. In GMShap, the $x_3$, which is past due within one month, also possesses a high weight.

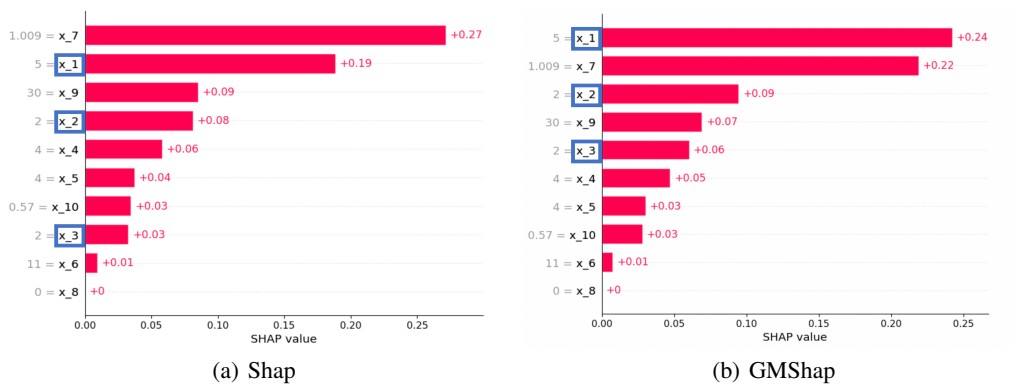

(a) Shap                 (b) GMShap

Figure 1: (CREDIT SCORING) Instance explanations by Shap and GMShap

We then examine strong pairwise monotonic features $x_1 - x_3$. We observe that Shap violates ASPM. Specifically, the average Shap is $[0.038 \quad 0.041 \quad 0.016]$, while the average GMShap is $[0.048 \quad 0.047 \quad 0.030]$. Consequently, Shap suggests that on average, each extended period of late payment is subject to fewer penalties than a short period of late payment. A misleading explanation such as this could result in negative consequences. According to this explanation, clients may believe that a longer delay will not adversely affect their credit scores and may even delay their future payments. Alternatively, GMShap preserves ASPM and sends a clear message that delays will negatively impact credit scores. A comparison at the global scale is provided in Appendix A.4.

## 6.2 RECIDIVISM - COMPAS

COMPAS is a scoring system that was developed to predict recidivism risk, which has been criticized for its racial bias by Angwin et al. (2016); Dressel & Farid (2018); Tan et al. (2018). Race and gender injustice have been extensively studied in the past by Foulds et al. (2020); Kearns et al. (2019; 2018); Hardt et al. (2016). The focus of our investigation is on the potential injustice associated with various types of offenses. Specifically, a felony is considered more serious than a misdemeanor. Without loss of generality, assume $x_1$ counts the number of felonies and $x_2$ counts the number of past misdemeanors. The probability of recidivism is strongly monotonic with respect to $x_1$ over $x_2$.

We examine the proportion of violations of strong pairwise monotonic features using Shap in this example. We limit ourselves to samples with potential violations (different numbers of felonies

---

[1] https://www.kaggle.com/c/GiveMeSomeCredit/overview

and misdemeanors that are both greater than zero), there are 46 data points, and nine of these, or 19.57%, violate ASPM. According to Shap, people may believe that a felony carries less seriousness than a misdemeanor, resulting in false perceptions. As opposed to this, GMShap clearly states that felonies are always considered more serious than misdemeanors. It is evident that GMShap should be adopted over Shap in this example.

## 6.3 FRAUD DETECTION - TWITTER BOTS ACCOUNTS

**Table 1** Average Shap and GMShap for $x_1$ and $x_2$. Violations by Shap are highlighted in purple.

| avgShap $x_1$ / avgShap $x_2$ | $x_1 = 100$ | $x_1 = 200$ | $x_1 = 300$ | $x_1 = 400$ |
|---|---|---|---|---|
| $x_2 = 100$ | 0.0056 / 0.00075 | 0.0028 / 0.00074 | 0.0019 / 0.00074 | 0.0014 / 0.00074 |
| $x_2 = 200$ | 0.0038 / 0.0012 | 0.0019 / 0.0012 | 0.0013 / 0.0012 | **0.00096** / **0.0012** |
| $x_2 = 300$ | 0.0032 / 0.0010 | 0.0016 / 0.0010 | 0.0011 / 0.0010 | **0.00081** / **0.0010** |
| $x_2 = 400$ | 0.0032 / 0.00079 | 0.0016 / 0.00079 | 0.0011 / 0.00079 | 0.00080 / 0.00079 |

| avgGMShap $x_1$ / avgGMShap $x_2$ | $x_1 = 100$ | $x_1 = 200$ | $x_1 = 300$ | $x_1 = 400$ |
|---|---|---|---|---|
| $x_2 = 100$ | 0.0038 / 0.0025 | 0.0021 / 0.0020 | 0.0016 / 0.0016 | 0.0013 / 0.0013 |
| $x_2 = 200$ | 0.0022 / 0.0020 | 0.0016 / 0.0016 | 0.0013 / 0.0013 | 0.0011 / 0.0011 |
| $x_2 = 300$ | 0.0016 / 0.0016 | 0.0013 / 0.0013 | 0.0011 / 0.0011 | 0.0009 / 0.0009 |
| $x_2 = 400$ | 0.0013 / 0.0013 | 0.0011 / 0.0011 | 0.0009 / 0.0009 | 0.0008 / 0.0008 |

The Twitter Bots Accounts dataset [2] is concerned with the detection of robot accounts on Twitter. We are primarily interested in the number of followers and friends in this dataset. According to Twitter, friends indicate that both accounts are being followed by each other, whereas followers indicate only one direction of following. Thus, the number of friends is a stronger indication that the account is not a robot. The probability of non-fraud is strongly monotonic with respect to the number of friends over the number of followers. For simplicity, we assume that $x_1$ counts the number of friends and $x_2$ counts the number of followers.

Taking advantage of the transparency of the MGNAM, we examine the results of Shap and GMShap at all possible values. Specifically, Shap and GMShap are applied to the output of the neural network $f_{1,2}(x_1, x_2)$ for variables $x_1$ and $x_2$. To check the preservation of ASPM, we calculate average (GM)Shap and we provide results for $100 \leq x_1, x_2 < 500$ for demonstration in Table 6.3. Shap violates ASPM in two parts, which are highlighted in purple. According to Shap, individuals may believe that the number of friends on average is a more reliable indicator of a legitimate account. In this way, if an individual's account is being questioned, he or she may unfollow some accounts in an attempt to improve their credibility, which is absurd.

## 7 CONCLUSION AND DISCUSSION

In this paper, we propose a new version of Shapley value to provide fair and reliable explanations for monotonic models. Based on our results, Shapley value may misinterpret domain knowledge. Therefore, **we must carefully investigate domain knowledge when explaining machine learning models**, especially for high-stakes sectors.

The monotonicity is studied in this work; however, there is numerous other domain knowledge that has not been studied (see for e.g., Gupta et al., 2020). It will be interesting to see how Shapley values work with other domain knowledge and whether our results can be generalized in the future.

---

[2] https://www.kaggle.com/datasets/davidmartngutirrez/twitter-bots-accounts

## 8  REPRODUCIBILITY STATEMENT

We have provided proofs for all theoretical results in Appendix A.1, A.2. We have also provided experimental details in Appendix A.4. Furthermore, we will release the code when the paper is accepted.

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

## A  APPENDIX

### A.1  MONOTONIC AXIOMS AND EXAMPLES

#### A.1.1  PROOFS

*Proof of Theorem 3.6.* Suppose we want to explain $\mathbf{x}^*$, let $\mathbf{x} = \mathbf{x}'$ in the definition of DIM, then it follows from the definition.

$\square$

**Lemma A.1.** *BShap preserves DIM.*

*Proof.* Here, we present a proof without cost-sharing assumptions, which is assumed in Friedman & Moulin (1999). Without loss of generality, suppose $f$ is individually monotonic with respect to $x_1$. Suppose $\mathbf{x} = (x_1, \mathbf{x}_\neg)$ and $\mathbf{x}^* = (x_1 + c, \mathbf{x}_\neg)$ for $c > 0$. Then

$$\mathrm{BS}_1(\mathbf{x}^*, \mathbf{x}', f) - \mathrm{BS}_1(\mathbf{x}, \mathbf{x}', f) = \sum_{S \subseteq N \setminus 1} \frac{|S|!(|N| - |S| - 1)!}{|N|!} \Delta f_S,$$

where

$$\Delta f_S = f(x_1 + c; \mathbf{x}_S, \mathbf{x}'_{N \setminus (S \cup 1)}) - f(x_1; \mathbf{x}_S, \mathbf{x}'_{N \setminus (S \cup 1)}) \geq 0$$

because of individual monotonicity. Thus, we conclude. $\square$

**Lemma A.2.** *For $x'_\beta = x'_\gamma$, BShap preserves AWPM.*

*Proof.* For simplicity, we prove the result for $\mathbf{x}' = 0$. Without loss of generality, suppose $f$ is weakly monotonic with respect to $x_1$ over $x_2$. For $\mathbf{x}' = \mathbf{0}$, we have $\frac{1}{x_1 - x'_1} = \frac{1}{x_2 - x'_2}$. Recall that

$$\mathrm{BS}_1 = \sum_{S \subseteq N \setminus 1} \frac{|S|!(|N| - |S| - 1)!}{|N|!} (v(S \cup 1) - v(S)).$$

For $\mathrm{BS}_2$, we use the symmetry, for each $S$ here, we consider $S'$ such that $x_1$ and $x_2$ are swapped within $S$, and everything else is left unchanged. That is, if $2 \notin S$, $S' = S$; if $2 \in S$, then $S' = (S \setminus 2) \cup 1$. For this arrangement, we only need to show that $v(S \cup 1) - v(S) \geq v(S' \cup 2) - v(S')$ in the summation. If $2 \notin S$, we have

$$v(S \cup 1) - v(S) = f(x_1, 0; \mathbf{x}_S, \mathbf{x}'_{N \setminus (S \cup \{1,2\})}) - f(0, 0; \mathbf{x}_S, \mathbf{x}'_{N \setminus (S \cup \{1,2\})}),$$
$$v(S' \cup 2) - v(S') = f(0, x_2; \mathbf{x}_{S'}, \mathbf{x}'_{N \setminus (S' \cup \{1,2\})}) - f(0, 0; \mathbf{x}_{S'}, \mathbf{x}'_{N \setminus (S' \cup \{1,2\})}).$$

Since $S = S'$ and $x_1 = x_2$, by the definition of weak pairwise monotonicity, we have

$$f(x_1, 0; \mathbf{x}_S, \mathbf{x}'_{N \setminus (S \cup \{1,2\})}) \geq f(0, x_2; \mathbf{x}_{S'}, \mathbf{x}'_{N \setminus (S' \cup \{1,2\})}),$$

and therefore $v(S \cup 1) - v(S) \geq v(S' \cup 2) - v(S')$.

If $2 \in S$, and we have

$$v(S \cup 1) - v(S) = f(x_1, x_2; \mathbf{x}_{S \setminus 2}, \mathbf{x}'_{N \setminus (S \cup 1)}) - f(0, x_2; \mathbf{x}_{S \setminus 2}, \mathbf{x}'_{N \setminus (S \cup 1)}),$$
$$v(S' \cup 2) - v(S') = f(x_1, x_2; \mathbf{x}_{S' \setminus 1}, \mathbf{x}'_{N \setminus (S' \cup 2)}) - f(x_1, 0; \mathbf{x}_{S' \setminus 1}, \mathbf{x}'_{N \setminus (S' \cup 2)}).$$

Since $S \setminus 2 = S' \setminus 1$, $S \cup 1 = S' \cup 2$, and $x_1 = x_2$, by the definition of weak pairwise monotonicity, we have

$$f(0, x_2; \mathbf{x}_{S \setminus 2}, \mathbf{x}'_{N \setminus (S \cup 1)}) \leq f(x_1, 0; \mathbf{x}_{S' \setminus 1}, \mathbf{x}'_{N \setminus (S' \cup 2)}),$$

and therefore $v(S \cup 1) - v(S) \geq v(S' \cup 2) - v(S')$.

Since $v(S \cup 1) - v(S) \geq v(S' \cup 2) - v(S')$ for all $S$ with corresponding $S'$, we conclude that $\mathrm{BS}_1 \geq \mathrm{BS}_2$.

$\square$

*Proof of Theorem 3.5.* The proof is followed by Theorem 3.6, Lemma A.1, and A.2. $\square$

*Proof of Theorem 3.4.* Suppose $f$ is individually monotonic with respect to $x_\alpha$, then

$$\mathrm{IG}_\alpha = (x_\alpha - x'_\alpha) \int_0^1 \frac{\partial f}{\partial x_\alpha} (\mathbf{x}' + t(\mathbf{x} - \mathbf{x}')) \, dt \geq 0,$$

since $x_\alpha \geq x'_\alpha$ and $\frac{\partial f}{\partial x_\alpha} \geq 0$.

Suppose $f$ is weakly monotonic with respect to $x_\beta$ over $x_\gamma$. For $x'_\beta = x'_\gamma$, we have $\frac{1}{x_\beta - x'_\beta} = \frac{1}{x_\gamma - x'_\gamma}$. Since $x_\beta = x_\gamma$ in $\mathbf{x}$ and $\mathbf{x}'$, $x_\beta = x_\gamma$ for $\mathbf{x}' + t(\mathbf{x} - \mathbf{x}')$, $\forall t \in [0, 1]$. $f$ is weakly monotonic respect to $x_\beta$ over $x_\gamma$, therefore $\frac{\partial f}{\partial x_\beta}(\mathbf{x}) \geq \frac{\partial f}{\partial x_\gamma}(\mathbf{x})$ if $x_\beta = x_\gamma$ in $\mathbf{x}$. Hence,

$$
\begin{aligned}
\text{IG}_\beta &= (x_\beta - x'_\beta) \int_0^1 \frac{\partial f}{\partial x_\beta} \left(\mathbf{x}' + t(\mathbf{x} - \mathbf{x}')\right) \, dt \\
&\geq (x_\beta - x'_\beta) \int_0^1 \frac{\partial f}{\partial x_\gamma} \left(\mathbf{x}' + t(\mathbf{x} - \mathbf{x}')\right) \, dt \\
&= (x_\gamma - x'_\gamma) \int_0^1 \frac{\partial f}{\partial x_\gamma} \left(\mathbf{x}' + t(\mathbf{x} - \mathbf{x}')\right) \, dt \\
&= \text{IG}_\gamma.
\end{aligned}
$$

Suppose $f$ is strongly monotonic with respect to $x_\beta$ over $x_\gamma$, then $\frac{\partial f}{\partial x_\gamma}(\mathbf{x}) \leq \frac{\partial f}{\partial x_\beta}(\mathbf{x})$, $\forall \mathbf{x} \in [\mathbf{a}, \mathbf{b}]$. Hence,

$$
\begin{aligned}
\frac{1}{x_\beta - x'_\beta} \text{IG}_\beta &= \int_0^1 \frac{\partial f}{\partial x_\beta} \left(\mathbf{x}' + t(\mathbf{x} - \mathbf{x}')\right) \, dt \\
&\geq \int_0^1 \frac{\partial f}{\partial x_\gamma} \left(\mathbf{x}' + t(\mathbf{x} - \mathbf{x}')\right) \, dt \\
&= \frac{1}{x_\gamma - x'_\gamma} \text{IG}_\gamma.
\end{aligned}
$$

$\square$

### A.1.2 EXAMPLES

**Example A.3** (An example of weak pairwise monotonicity). *An applicant who intends to study STEM in graduate school is required to take the GRE general (170 in math and 170 in verbal) as one of the factors for admission. Assume $f$ is the admitted probability, $x_\beta$ is the student's math score, and $x_\gamma$ is the student's verbal score. Due to the importance of math in STEM, $f$ should be weakly monotonic with respect to $x_\beta$ over $x_\gamma$. In this case, it is necessary to fulfill the condition $x_\beta = x_\gamma$. If the student has the same level of math and verbal skills, it is more desirable to see improvement in math. It is a different story when a student has a strong math score but a weak verbal score. There is often a requirement for a minimum verbal score in schools in order to ensure effective communication. A student who has a strong math score but a very weak verbal score may be able to improve his/her chances of admission more significantly if he/she improves his/her verbal score.*

**Example A.4** (Failure of IG for DIM). *The following example in Friedman & Moulin (1999) provides a counterexample with a comparison between IG and BShap.*

*Consider the function $f(x_1, x_2) = \frac{x_1 x_2}{x_1 + x_2}$ with baseline $\mathbf{x}' = (0, 0)$. For IG, it can be shown that $IG_1(\mathbf{x}, \mathbf{x}', f) = \frac{x_1 x_2^2}{(x_1 + x_2)^2}$. Note $\frac{\partial}{\partial x_1} IG_1 = \frac{x_2^2(x_2 - x_1)}{(x_1 + x_2)^3}$, therefore DIM is not preserved. It is important to note that the path for the new explicand has been changed. As a result, the new path cannot guarantee greater attributions for the explicand.*

*For BShap, we have $BS_1(\mathbf{x}, \mathbf{x}', f) = \frac{x_1 x_2}{2(x_1 + x_2)}$, which preserves DIM. The main difference is that in the BShap calculation, the new path covers the old path.*

**Example A.5** (Impact of baseline point in the AWPM). *Results of AWPM axioms also suggest baseline points. We need $x'_\beta = x'_\gamma$ in $\mathbf{x}'$ in order to preserve AWPM.*

*Consider $f(x_1, x_2) = 4.5x_1 - x_1^2 + 4x_2 - x_2^2$ for $x_1, x_2$ in $[0, 2]$. $f$ is weakly monotonic with respect to $x_1$ over $x_2$. Consider the baseline $\mathbf{x}' = (0, 0)$ and the explicand $\mathbf{x} = (2, 2)$.*

*For IG, we have*

$$
IG((2, 2), (0, 0), f) = \begin{bmatrix} 5 \\ 4 \end{bmatrix}.
$$

*It can be seen that $\frac{1}{2}IG_1 > \frac{1}{2}IG_2$, IG preserves AWPM. Now suppose we consider $\mathbf{x}' = (1, 0)$, we have*

$$IG((2, 2), (1, 0), f) = \begin{bmatrix} 1.5 \\ 4 \end{bmatrix}.$$

*Since $IG_1 < \frac{1}{2}IG_2$, IG fails to preserve AWPM. The result of BShap is the same as that of $f$ being additively separable.*

*Consider Example A.3. For $x'_\beta = x'_\gamma = 0$, the weak pairwise monotonicity in $f$ guarantees that math is more important than verbal when they are equal when we determine the general importance of features. If, on the other hand, we take the average historical statistics of admitted students as our baseline, then it would be possible to see that verbal skills are more important than mathematics skills as admitted students are already competent in mathematics. It is important to keep in mind in this case that we are comparing the cases of previously admitted students, and the importance of features does not necessarily apply in general.*

**Example A.6** (Failure of IG for discrete features). *Consider a two-dimensional set function $f$ with $f(0, 0) = f(1, 0) = f(0, 1) = 0, f(1, 1) = 1$. Let us suppose that two continuous functions are used as the approximation and that we have*

$$g_1(x, y) = xy, \quad g_2(x, y) = x^{99}y. \tag{25}$$

*Both functions perfectly match $f$, but IG could produce completely different results:*

$$IG(\mathbf{1}, \mathbf{0}, g_1) = \begin{bmatrix} 0.5 \\ 0.5 \end{bmatrix}, \quad IG(\mathbf{1}, \mathbf{0}, g_2) = \begin{bmatrix} \frac{99}{100} \\ \frac{1}{100} \end{bmatrix}. \tag{26}$$

*As a result, IG violates implementation invariance when discrete features are involved.*

**Example A.7** (Is DIM always required?). *Consider Example 2.6 with a concrete example, where $f(x_1, x_2) = \min(0.2x_1 + 0.1x_2, 0.3)$ and $x_1$ is strongly monotonic with respect to $x_2$. At $\mathbf{x} = (1, 1)$ with baseline $\mathbf{x}' = (0, 0)$, the corresponding attribution should be $\mathbf{A}(\mathbf{x}, \mathbf{x}', f) = (0.2, 0.1)$, due to linearity. Now we consider $\mathbf{x}^* = (3, 1)$, $f(\mathbf{x}^*) = 0.3$ and DIM requires that $A_1(\mathbf{x}^*, \mathbf{x}', f) \geq 0.2$. In this case, we wish to satisfy this requirement. However, if we consider $\mathbf{x}^! = (0, 4)$ with $f(\mathbf{x}^!) = 0.3$. From the perspective of $\mathbf{x}^!$, $x_1$ does not incur an additional cost. Accordingly, a unit of $x_1$ and $x_2$ should be attributed equally. This results in attributions at $(3, 1)$ of $(0.15, 0.15)$, which violates the principle of DIM.*

### A.2 PROOFS IN STRONG MONOTONIC GAMES

**Lemma A.8.** *MShap preserves implementation invariance.*

*Proof.* By Definition. $\qquad\square$

**Lemma A.9.** *MShap preserves linearity.*

*Proof.* Suppose $f = af_1 + bf_2$, then by definition, we have $w_i(\mathbf{x}, f) = aw_i(\mathbf{x}, f_1) + bw_i(\mathbf{x}, f_2)$. For $\sum_{j=1}^i x_i = 0$, $\phi_i(f) = 0 = a\phi_i(f_1) + b\phi_i(f_2)$. For $\sum_{j=1}^i x_i \neq 0$, we have

$$\begin{aligned} \phi_i(f) &= x_i \sum_{j=i}^m \frac{a(w_j(f_1) - w_{j+1}(f_1)) + b(w_j(f_2) - w_{j+1}(f_2))}{\sum_{k=1}^j x_k} \\ &= a\phi_i(f_1) + b\phi_i(f_2). \end{aligned}$$

$\qquad\square$

**Lemma A.10.** *MShap is complete.*

*Proof.* For $x_1 \neq 0$. For each $j$, we sum the coefficients of $w_j$. In the case of $j = 1$, the only coefficient that is nonzero is $x_1 \frac{1}{x_1} = 1$. Next, we consider the case in which $j > 1$. In the case of $\phi_i$ with $i < j$, we have nonzero coefficients $x_i \left( -\frac{1}{\sum_{k=1}^{j-1} x_k} + \frac{1}{\sum_{k=1}^j x_k} \right)$. In the case of $\phi_i$ with

$i = j$, we have nonzero coefficients $x_i \frac{1}{\sum_{k=1}^{j} x_k}$. The calculation of $\phi_i$ with $i > j$ doesn't depend on $j$. Take the summation, we have

$$\sum_{i=1}^{j-1} x_i \left( -\frac{1}{\sum_{k=1}^{j-1} x_k} + \frac{1}{\sum_{k=1}^{j} x_k} \right) + x_j \frac{1}{\sum_{k=1}^{j} x_k} = -1 + 1 = 0.$$

For $x_1 = 0$ but $x_2 \neq 0$, by Lemma A.13, we have $\phi_1 = 0$. With the same argument above, $\sum_{j=2}^{m} \phi_i = f(0, x_1 + x_2, x_3, \ldots, m) = f(\mathbf{x})$. Extend this argument to $\sum_{j=1}^{i} x_j = 0$ and $x_{i+1} \neq 0$ for $i = 1, \ldots, m - 1$, we conclude. $\qquad \square$

**Lemma A.11.** *MShap preserves average individual monotonicity.*

*Proof.* If $x_i = 0$ or $\sum_{j=1}^{i} x_j = 0$, then $\phi_i = 0$. Otherwise, $w_i \geq w_{i+1}, i = 1, \ldots, m - 1$ due to strong pairwise monotonicity and $w_m \geq 0$ due to individual monotonicity for $x_m$. Therefore, we conclude. $\qquad \square$

**Lemma A.12.** *MShap preserves average strong pairwise monotonicity.*

*Proof.* Suppose $i > j$ and $x_i, x_j > 0$. Then by strong pairwise monotonicity, we have

$$\frac{1}{x_i} \phi_i - \frac{1}{x_j} \phi_j = \sum_{k=i}^{j-1} \frac{w_k(\mathbf{x}) - w_{k+1}(\mathbf{x})}{\sum_{l=1}^{k} x_l} \geq 0.$$

$\qquad \square$

*Proof of Lemma 4.2.* The proof is followed by Lemma A.8, A.13, A.9, A.10, A.11, and A.12. $\qquad \square$

**Lemma A.13.** *MShap preserves dummy(b).*

*Proof.* Suppose $x_i > 0$ is a dummy, then

$$f(0, \ldots, 0, x_i, 0, \ldots, 0) = f(\mathbf{0}) = 0, \ \forall x_i.$$

Because of strong pairwise monotonicity, we have

$$f(0, \ldots, x_i, \ldots, x_m) \leq f\left( 0, \ldots, \sum_{j=i}^{m} x_j, \ldots, 0 \right) = 0.$$

Therefore, for $j \geq i$, $w_j = 0$. Thus, $\phi_i = 0$.

Suppose $x_i = 0$, we have $\phi_i = 0$ by formula. In addition, for $j > i$, $\phi_i$ doesn't involve $w_i$. For $j < i$, we have

$$\sum_{j=1}^{i-1} x_j = \sum_{j=1}^{i} x_j.$$

Therefore,

$$\frac{w_{i-1}(\mathbf{x}) - w_i(\mathbf{x})}{\sum_{j=1}^{i-1} x_j} + \frac{w_i(\mathbf{x}) - w_{i+1}(\mathbf{x})}{\sum_{j=1}^{i} x_j} = \frac{w_{i-1}(\mathbf{x}) - w_{i+1}(\mathbf{x})}{\sum_{j=1}^{i-1} x_j},$$

which is not dependent on $w_i$. Thus, we conclude. $\qquad \square$

**Lemma A.14.** *MShap preserves symmetry(b).*

*Proof.* By strong pairwise monotonicity, we know

$$f\left(x_1, \ldots, x_{k-1}, \sum_{i=k}^{l} x_i, 0, \ldots, 0, x_{l+1}, \ldots, x_m\right)$$
$$\geq f(x_1, \ldots, x_m)$$
$$\geq f\left(x_1, \ldots, x_{k-1}, 0, \ldots, \sum_{i=k}^{l} x_i, x_{l+1}, \ldots, x_m\right).$$

Since $f$ is symmetric about $x_k$ and $x_l$, $f(\mathbf{x}) = f(\mathbf{x}^*)$ where $x_i = x_i^*$ for $i = 1, \ldots, k-1$ and $l+1, \ldots, m$, and $\sum_{i=k}^{l} x_i = \sum_{i=k}^{l} x_i^*$ for all $\mathbf{x}$ and $\mathbf{x}^*$. Then $w_i = w_j$ for $k \leq i, j \leq l$. Therefore, for $k \leq i \leq l$, we have

$$\phi_i(\mathbf{x}) = x_i \sum_{j=l}^{m} \frac{w_j(\mathbf{x}) - w_{j+1}(\mathbf{x})}{\sum_{k=1}^{j} x_k}.$$

Thus, we conclude. $\qquad\square$

*Proof of Lemma 4.5.* The result is followed from Lemma A.13 and A.14. $\qquad\square$

*Proof of Lemma 4.6.* Suppose $\mathbf{x} = (x_1, \ldots, x_m)$ and $\mathbf{x}^* = (x_1, \ldots, x_m + c)$, where $c > 0$. We calculate

$$\phi_m^* = x_m^* \frac{w_m(\mathbf{x}^*)}{\sum_{j=1}^{m} x_j^*} = (x_m + c) \frac{f(0, \ldots, \sum_{j=1}^{m} x_j + c)}{\sum_{j=1}^{m} x_j + c} \geq x_m \frac{f(0, \ldots, \sum_{j=1}^{m} x_j)}{\sum_{j=1}^{m} x_j} = \phi_m,$$

where $\frac{x_m + c}{\sum_{j=1}^{m} x_j + c} = 1 - \frac{\sum_{j=1}^{m-1} x_j}{\sum_{j=1}^{m} x_j + c} \geq 1 - \frac{\sum_{j=1}^{m-1} x_j}{\sum_{j=1}^{m} x_j} = \frac{x_m}{\sum_{j=1}^{m} x_j}$ and $f(0, \ldots, \sum_{j=1}^{m} x_j + c) \geq f(0, \ldots, \sum_{j=1}^{m} x_j)$ due to individual monotonicity. $\qquad\square$

*Proof of Theorem 4.7.* We prove this by induction. We want to construct a matrix $\mathbf{A}$ such that $\mathbf{A}\mathbf{w} = \phi$. From Lemma 4.2 and Lemma A.14, we already know MShap satisfies all these axioms. Now we want to show that $\phi$ is unique. For $\mathbf{x} > \mathbf{0}$, we want to show that $\mathbf{A}$ is unique, and here are the requirements that $\mathbf{A}$ needs to satisfy.

1. By dummy(b) and ASPM, $\mathbf{A}$ is an upper triangular matrix.

2. By completeness, dummy(b), and ASPM, we know $\sum_i A_{i,j} = 0$ for all $j > 1$ and $A_{1,1} = 1$.

3. By symmetry(b), ASPM and completeness, if $f$ is symmetric about $x_1$ and $x_m$, then we require that $\phi_i = \frac{x_i}{\sum_{i=1}^{m} x_i} f(\mathbf{x})$. This implies that

$$\mathbf{A}\mathbf{1} = \frac{1}{\sum_{i=1}^{m} x_i} \mathbf{x}.$$

We start with the case two features. By satisfying Conditions 1 and 2, we have

$$\mathbf{A} = \begin{bmatrix} 1 & a \\ 0 & 1 - a \end{bmatrix}.$$

By satisfying Condition 3, we have

$$\mathbf{A} = \begin{bmatrix} 1 & \frac{x_1}{x_1 + x_2} - 1 \\ 0 & \frac{x_2}{x_1 + x_2} \end{bmatrix}.$$

Therefore, $\mathbf{A} \in \mathbb{R}^{2 \times 2}$ is unique. Now we show it by induction. Suppose $\mathbf{A}_m$ is uniquely defined for $m$, we show it is also uniquely defined for $m + 1$. Suppose $x_{m+1}$ is a dummy, then we should obtain the exact same formula for $\phi_i$, $i = 1, \ldots, m$. Therefore, we must have

$$\mathbf{A}_{m+1} = \begin{bmatrix} \mathbf{A}_m \in \mathbb{R}^{m \times m} & \mathbf{b} \in \mathbb{R}^{m \times 1} \\ \mathbf{0} \in \mathbb{R}^{1 \times m} & c \in \mathbb{R}^{1 \times 1} \end{bmatrix}.$$

Now by Condition 3, the last column is uniquely determined as

$$\begin{bmatrix} \mathbf{b} \\ c \end{bmatrix} = \frac{1}{\sum_{i=1}^{m} x_i} \begin{bmatrix} \mathbf{x}_m \\ x_{m+1} \end{bmatrix} - \begin{bmatrix} \mathbf{A}_m \mathbf{1}_m \\ \mathbf{0} \end{bmatrix},$$

which is the same as the Definition 4.1.

Now suppose $x_i = 0$ and $x_j \neq 0$ for $j \neq i$, by dummy(b), we know $\phi_i = 0$ and $\phi_j$ for $j \neq i$ can be determined by calculating $A_j(f(x_1, \ldots, x_{i-1}, 0, x_{i+1}, \ldots, x_m))$, which we just show its uniqueness. Continuing this argument, we finish the proof. $\qquad\square$

### A.3 GMSHAP EXAMPLE

**Example A.15.** *Suppose we have $N = \{\{1, 2\}, 3\}$, then the characteristic function $v$ only consider $v(\emptyset)$, $v(12)$, $v(3)$, and $v(123)$.*

*For dummy, we say $\{1, 2\}$ is a dummy player if*

$$v(\{1, 2\}) = 0,$$
$$v(\{1, 2, 3\}) - v(\{3\}) = 0.$$

*For symmetry, $\{1, 2\}$ and $\{3\}$ are symmetric if*

$$v(\{1, 2\}) = v(\{3\}),$$
$$v(\{1, 2, 3\}) - v(\{3\}) = v(\{1, 2, 3\}) - v(\{1, 2\}).$$

**Example A.16.** *Suppose we have $N = \{\{1, 2\}, 3\}$, where $f$ is strongly monotonic with respect to $x_1$ over $x_2$. Consider*

$$f = \log(1 + 10x_1 + 9x_2 + x_3).$$

*and we are interested in the explicand $\mathbf{x} = (4, 1, 2)$ with the baseline $\mathbf{x}' = (0, 0, 0)$. First, we calculate GShap values. By calculation, we have*

$$v(\{3\}) = \log(3), \ v(\{1, 2\}) = \log(50), \ v(\{1, 2, 3\}) = \log(52).$$

*GShap values are calculated by*

$$\Phi_1 = \frac{v(\{1, 2\})}{2} + \frac{v(\{1, 2, 3\}) - v(\{3\})}{2} \approx 3.38,$$
$$\Phi_2 = \frac{v(\{3\})}{2} + \frac{v(\{1, 2, 3\}) - v(\{1, 2\})}{2} \approx 0.57.$$

*The further calculation of GMShap yields that*

$$\Phi_{1,1} = \frac{1}{2}\left( (f(4, 1, 0) - f(0, 5, 0)) + \frac{4}{5} f(0, 5, 0) \right)$$
$$+ \frac{1}{2}\left( (f(4, 1, 2) - f(0, 5, 2)) + \frac{4}{5}(f(0, 5, 2) - f(0, 0, 2)) \right) \approx 2.72,$$
$$\Phi_{1,2} = \frac{1}{2} \cdot \frac{1}{5} \cdot f(0, 5, 0) + \frac{1}{2} \cdot \frac{1}{5} \cdot (f(0, 5, 2) - f(0, 0, 2)) \approx 0.66.$$

### A.4 DATA AND MODEL

We utilize the monotonic groves of neural additive models (MGNAM) by Chen & Ye (2023), which is a relatively transparent model that enforces pairwise monotonicity. In general, the model has the form

$$g(\mathbb{E}[y|\mathbf{x}]) = f(\mathbf{x}),$$

where $g$ is the link function (e.g., $g^{-1}$ is the logistic function for classifications). According to the feature properties, $f(\mathbf{x})$ will be further decomposed. For each example, details will be provided. The choice of the model is not unique. Models developed in Liu et al. (2020); Milani Fard et al.

(2016); You et al. (2017); Runje & Shankaranarayana (2023) are applicable for individual mono-tonicity, whereas deep lattice models (Gupta et al., 2020; Cotter et al., 2019) include strong pairwise monotonicity. The architecture of the model is one hidden layer for each neural network with four hidden neurons. Additionally, all activation functions are set as sigmoid. As an illustration, we focus on simple architectures. It is possible to improve the performance of the model, but it is not the focus of this study. When checking for accuracy, the dataset is randomly partitioned into $70\%$ training and $30\%$ test sets. The baseline point is chosen $\mathbf{x}' = \mathbf{0}$. It is also not a unique choice and can be modified.

### A.4.1 CREDIT SCORING - GIVE ME SOME CREDITS

- $x_1 - x_3$: Last two years, the number of times the borrower was 90+ days past due, 60-89 days past due, and 30-59 days past due.
- $x_4$: Monthly income.
- $x_5$: Number of dependents in the family.
- $x_6$: Number of open loans and lines of credit.
- $x_7$: Total balance on credit cards and personal lines of credit except for real estate and no installment debt such as car loans divided by the sum of credit limits.
- $x_8$: Number of mortgage and real estate loans.
- $x_9$: Age of borrower in years.
- $x_{10}$: Monthly debt payments, alimony, and living costs divided by monthly gross income.
- $y$: Client's behavior; 1 = Person experienced 90 days past due delinquency or worse.

For simplicity, data with missing variables are removed. Past dues that are greater or equal to 20 are discarded as they either represent missing information or outliers. Then past dues greater than five times are replaced by five due to the rarity. This also applies to $x_5$ if its value exceeds five.

For MGNAM, we consider the architecture

$$f(\mathbf{x}) = f_{1,2,3}(x_1, x_2, x_3) + f_4(x_4) + \cdots + f_{10}(x_{10}).$$

In other words, $x_1 - x_3$ are grouped together, and the remaining features are handled using 1-dimensional functions. For $x_1 - x_3$, we enforce strong pairwise monotonicity. We enforce individual monotonicity for $x_4 - x_5$. The area-under-the-curve (AUC) of the model is around $85\%$, which indicates that the model is accurate.

In addition, Shap and GMShap values are compared at a global level. In particular, we focus on samples that have at least two values in $[x_1, x_2, x_3]$ that are greater than zero, because GMShap coincides with Shap for the remainder of the samples. Also, we exclude the case where $x_1 = x_2 = x_3$, since Shap will not violate this condition . The global explanation for the restricted samples is provided in Figure 2. Shap provides explanations that are fair on a global scale, which is similar to GMShap's global attributions.

### A.4.2 RECIDIVISM - COMPAS

COMPAS is a proprietary score developed to predict recidivism risk, which is used to guide bail, sentencing, and parole decisions. A report published by ProPublica in 2016 provided recidivism data for defendants in Broward County, Florida (Pro, 2016). We focus on the simplified cleaned dataset provided in Dressel & Farid (2018). Three thousand and fifty-one ($45\%$) of the 7,214 observations committed a crime within two years. This study uses a binary response variable, recidivism, as the response variable. The dataset here contains nine features selected after some feature selection was conducted.

- $x_1$: Total number of juvenile felony criminal charges
- $x_2$: Total number of juvenile misdemeanor criminal charges
- $x_3$: Age
- $x_4$: Total number of non-juvenile criminal charges

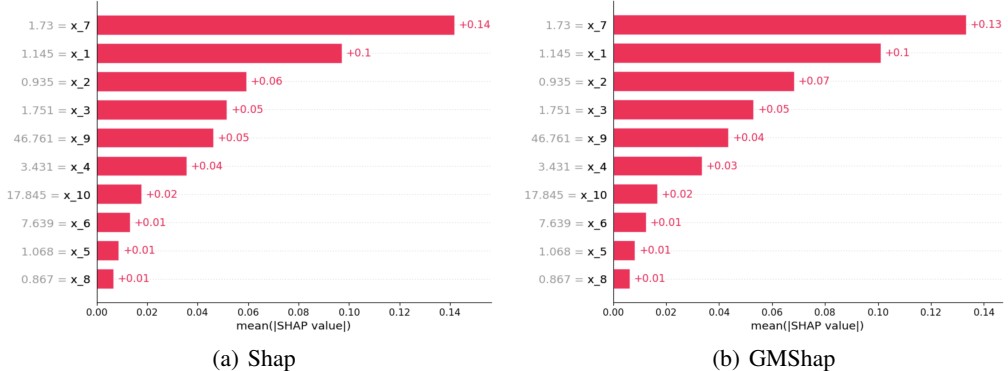

(a) Shap $\qquad$ (b) GMShap

Figure 2: (CREDIT SCORING) Global explanations for the restricted samples by Shap and GMShap

- $x_5$: A numeric value corresponding to the specific criminal charge
- $x_6$: An indicator of the degree of the charge: misdemeanor or felony
- $x_7$: Races include White (Caucasian), Black (African American), Hispanic, Asian, Native American, and Others
- $x_8$: Sex, male or female
- $x_9$: A numeric value between 1 and 10 corresponds to the recidivism risk score generated by COMPAS software (a small number corresponds to a low risk, and a larger number corresponds to a high risk)
- $y$: Whether the defendant recidivated two years after the previous charge

To avoid discrimination, we further exclude races and sexes. The COMPAS score is also excluded as it is not the focus of this study and is correlated with other features, making its interpretation more difficult. As there are too few samples, we truncate the number of juveniles exceeding five. Otherwise, if monotonicity is requested, neural network functions will become flat, which is not helpful.

For MGNAM, we consider the architecture

$$f(\mathbf{x}) = f_{1,2}(x_1, x_2) + f_3(x_3) + \cdots + f_6(x_6). \tag{27}$$

In other words, $x_1 - x_2$ are grouped, and the remaining features are handled using 1-dimensional functions. For $x_1 - x_2$, we enforce strong pairwise monotonicity. The AUC of the model is about 72%, which is consistent with the literature (Dressel & Farid, 2018).

### A.4.3 FRAUD DETECTION - TWITTER BOTS ACCOUNTS

Twitter Bots Accounts (Ramalingaiah et al., 2021; Shukla et al., 2021) is a classification dataset that seeks to categorize whether an account is operated by a human or a bot. Here is a demonstration of the features in the dataset.

- $x_1$: The number of friends accounts
- $x_2$: The number of followers accounts
- $x_3$: The default profile for which an account has or not
- $x_4$: The number of favorite accounts
- $x_5$: Whether the geological information is accessible or not
- $x_6$: A total count of tweets (including retweets) posted by a user
- $x_7$: Whether the account is verified or not
- $x_8$: The average online time per day

- $x_9$: The time length for which the account has been created
- $x_{10}$: Time the information created
- $x_{11}$: The default profile image for which an account has or not
- $x_{12}$: The description of an account
- $x_{13}$: The account's id
- $x_{14}$: The language setting of the account
- $x_{15}$: Where the account belongs (location)
- $x_{16}$: The URL of the profile background image
- $x_{17}$: The URL of the profile image
- $x_{18}$: The screen name that is shown on the Twitter interface
- $y$: Whether the account belongs to a 'bot' or 'human'

There is a strong pairwise monotonicity relation between the feature $x_1$ (the number of friends) and the feature $x_2$ (the number of followers). Due to the bidirectional nature of the relationship, the number of friends is more valuable than the number of followers. As a result, we impose pairwise monotonicity on both of these features in the MGNAM model. During feature cleaning, nonimportant information, such as $x_{10-18}$, is ignored. The boolean information of $x_3$, $x_5$, and $x_7$ is transformed into 0 and 1. The label 'bot' is transformed into 0 and the label 'human' into 1. For feature manipulation, we concate $x_1$ and $x_2$ greater than 5000. For better illustration, we discretized the data by partitioning it in intervals with size 100: $x \in [0, 100) = 0$, $x \in [100, 200) = 100$, $x \in [200, 300) = 200$, $x \in [300, 400) = 300$, .... Efforts can be made to improve these results, but that is not the primary focus of this study.

For MGNAM, we consider the architecture

$$f(\mathbf{x}) = f_{1,2}(x_1, x_2) + f_3(x_3) + \cdots + f_9(x_9). \tag{28}$$

In other words, $x_1 - x_2$ are grouped together, and the remaining features are handled using 1-dimensional functions. For $x_1 - x_2$, we enforce strong pairwise monotonicity. The AUC of the model is about 77%. In this example, Shap and GMShap are applied to the 2-dimensional output of a neural network, $f_{1,2}(x_1, x_2)$ for features $x_1$ and $x_2$.

