# OpenReview forum: "Fairly Explaining Monotonic Models: a New Shapley Value"
_ICLR.cc/2024/Conference — ICLR 2024 Conference Withdrawn Submission_

### Official Review · Reviewer_bmYC · 2023-10-27

**Soundness:** 2 fair
**Presentation:** 2 fair
**Contribution:** 1 poor
**Rating:** 3
**Confidence:** 3

**Summary:**

This paper studies the problem of explaining machine learning (ML) models, in particular their predictions, and focus on the monotonic models. The paper argues that the Shapley value-based method is ineffective in this case and proposes a variant called monotonic Shapley, in the strong monotonic games. Empirical results w.r.t. a specific class of monotonic ML models show that the proposed method is able to preserve monotonicity better than the Shapley value based method.

**Strengths:**

- The paper studies an important problem, of explaining (the predictions) of machine learning models.

- The theoretical treatment, including the derivations of the results, is appreciated.

- The application of the method to real-world datasets illustrates its applicability.

**Weaknesses:**

- The notion of science-informed ML models can be made clearer, better justified and utilized.

    - To elaborate, other than monotonicity, it seems there isn't another property of the science-informed ML models is formalized or used. Moreover, it is not entirely clear how or why monotonicity is applicable to science-informed ML models (e.g., physics-informed ML models that preserve the conservation of energy laws).
    - On the other hand, it may be arguable that the finance-based models (which the authors claim to require mononicty) fit well with the categorization of "science-informed models".

- Motivation of the setting, and subsequently definitions and approach can be made stronger and clearer.
    - Section 3.2 describes some axioms considered here. However, it can be better motivated why these axioms are needed for the considered setting here (i.e., science-informed ML models).
    - The monotonicity definitions seem to be adaptations of an existing work, and the motivation for why these definitions can be made stronger. For instance, what is the implication of a BAM satisfying ASPM?


- The restrictiveness of the required assumptions takes away the contribution of the work and the proposed method.

    - The assumption of knowing pairwise strong monotonicity of the function, and which features.

    - Application to a restricted class of ML models, namely MGNAMs, in order to satisfy such assumption. To elaborate, the authors of the method MAGAMs have remarked that "We aim to develop a new model that will maintain transparency to the greatest extent possible, in the manner of Neural additive models (NAMs) (Agarwal et al., 2021, Chen & Ye, 2022)."

        It is not so clear or convincing to illustrate the effectiveness of mSHAP to this class of already (somewhat) transparent and interpretable ML models, and yet mSHAP requires the assumption which is satisfied this class of models.

**Questions:**

1. In introduction,
    > These models can be categorized as science-informed machine learning models.

    While I agree that physics-informed ML models can be categorized as science-informed, can the finance-related ML models really be categorized as science-informed? Moreover, this characterization does not seem to be fully used: the physics-informed ML models are sometimes enforced to satisfy the conservations laws (as an example), but it is not mentioned or used in the rest of the work at all. On the other hand, the focus is mainly on monotonicity, which is primarily motivated by the finance applications mentioned here.

2. In introduction,
    > a fair credit scoring system should punish each additional late payment.

    Is there actual implemented pracice to justify this, or is it primarily hypothetical?

3.  In introduction,

    > when pairwise monotonicity is involved, Shap can often produce misleading explanations and produce unfair interpretations.

    Exactly, what does "unfairness" refer to here?

    > Fortunately, GMShap has avoided these issues and has been able to provide reasonable and reliable explanations.

    What do "reasonable and reliable" refer to? And how are they related to (un)fairness?

4. In Sec. 2.1

    > [a, b] to be the hyperrectangle.

    Do you mean the Cartesian product of a,b? How is this setting specifically used? In other words, what is the problem of assuming the input space is $R^n$?

5. In Equation (2), how is $x'_{N \setminus S}$ defined?

6. In Equation (3), should the arguments follow $f$ in the numerator?

7. In Sec. 2.3,

    > we list axioms that are considered in this paper.

    What is the motivation of considering all of the following axioms, specific to your setting of monotonic models, in particular completeness, linearity, dummy and symmetry?

8. In Equation (11), what is the value of $i$ in the definition of $x^*$?

9. In Sec. 4,

    > We further assume that $x_i \in \mathbb{R}^+, \forall i$

    How to satisfy this in practice? For instance, the common practice of feature processing is standardization and can lead to negative feature values.

**Details Of Ethics Concerns:**

N.A.

---

### Official Review · Reviewer_TM3d · 2023-10-28

**Soundness:** 1 poor
**Presentation:** 2 fair
**Contribution:** 2 fair
**Rating:** 3
**Confidence:** 3

**Summary:**

This paper proposes a new Shapley value for explaining the predictions of ML models. The authors argue that traditional Shapley values, which are widely used for explaining black-box models, do not always reflect desired properties like monotonicity, especially pairwise monotonicity. To address this issue, the authors introduce a new method that modifies the traditional Shapley value calculation to account for the monotonicity constraints of the model. The authors demonstrate the effectiveness of their method on several real-world datasets including credit scoring, recidivism, and fraud detection.

**Strengths:**

1. This paper studies an interesting question of the monotonicity of the Shapley value.

2. A nice review of the Baseline Attribution Method and Shapley value in the preliminary.

3. Experiments include important cases urgently needing XAI, i.e., credit scoring, recidivism, and fraud detection.

**Weaknesses:**

1. I have concerns about the motivation and the core definition of the pairwise monotonicity proposed in this paper. Please refer to my question 1 below.

2. The paper needs to state its scope and motivation better. The problem addressed is monotonicity, but the scope is set to science-informed ML and domain knowledge for XAI in the paper, especially in the introduction.

**Questions:**

1. The core property that motivates this paper is the pairwise monotonicity. However, I have concerns about why this is important and why the current definition in Def. 2.4 and 2.5 make sense.

For both Def. 2.4 and 2.5, the same value c is added to the features x_beta and x_gamma for monotonicity. However, this doesn't make much sense to me as the distribution of x_beta and x_gamma is unknown. Adding the same value to a feature with a large range vs. a feature with a small range is not fair. This unclear definition raises a further concern about in what cases will pairwise monotonicity be truly useful. Will it be useful to define it between any pair of features? Without clarifying this concern, it is hard to assess how valuable the rest of the analysis is.

---

### Official Review · Reviewer_Kdh1 · 2023-10-29

**Soundness:** 2 fair
**Presentation:** 2 fair
**Contribution:** 1 poor
**Rating:** 3
**Confidence:** 4

**Summary:**

The shapley value has been widely used as an attribution method for explaining black-box machine learning models. A rigorous mathematical framework based on a number of axioms has enabled Shapley value to disentangle the black-box structure of models. Recent studies have shown that domain knowledge is an important component of machine learning models. Science-informed machine learning models that incorporate domain knowledge have demonstrated better generalization and interpretation capabilities. But when the users apply attribution models to science-informed machine learning models, it's still unknown if they can obtain consistent scientific explanations. In this work, the authors show that Shapley value can not be guaranteed to reflect domain knowledge, such as monotonicity. To remedy Shapley's monotonicity failure, the authors propose a new version of Shapley value. As a result of extensive analytical and empirical results, they show that conventional Shapley value often produces misleading explanations for monotonic models, which can be addressed by the new method.

**Strengths:**

1. The writing of this paper is generally acceptable and not hard to follow.
2. The authors provide the detailed theoretical derivation process and specific examples in the Appendix.

**Weaknesses:**

1. In this paper, the authors focus on the pairwise monotonicity of the machine learning models. But the importance of this characeteristic in the general machine learning research is less elaborated, which can hardly convince others this is a meaningful work.
2. According to my understanding, the monotonicity, especially the pairwise monotonicity, is just one kind of conservation laws in the science-informed machine learning. Though the proposed GMShap in the paper is specially designed to address the pairwise monotonicity, can it be extended to other conservation laws, like the conservation law of momentum/energy/mass/charge/angular momentum/linear momentum. If not, the scope of this work has actully been greatly limited. Do we need to design a corresponding improved Shapley value version for each above physics conservation law?
3. The motivation of this paper should be questioned. In the introduction part, the authors claim that when the pairwise monotonicity is involved, Shap can often produce misleading explanations and unfair interpretations. The title of this paper is also **fairly explainng monotonic models**. However, in the Sec. 3.2 **Preservation and Failure of Axioms**, the authors only show that BShap may violate the Average Strong Pairwise Monotonicity (ASPM). However, why violating ASPW necessarily means the misleading explanations and unfair interpretations, especially considering that BShap can still preserve AIM, DIM and AWPM?
4. The strong monotonic games setting is not easy to meet in real applications. For example, the authors require that $f(x^{'})=0$ when $x^{'}=0$. However, in the most common machine learning models like neural networks, the bias term is almost necessary which makes the condition difficult to satisty. Thus, it should be questioned if this problem setting is over idealistic.
5. If the two-step generalized monotonic shapley value framework brings extra computation cost is still unknown. Considering that original Shapley value computation has been the computationally heavy task, the complexity of the newly proposed GMShap should be necessarily analyzed. Otherwise, its applicability will be strongly doubted.
6. The model used in the experiments lacks the enough representativeness. How about the performance of GMShap in other monotonous  machine learning models? More experiments on this point are expected.
7. Some descriptions and points in this paper should be further clarified.

**Questions:**

1. What does the *subgame* in Definition 5.1 mean?
2. I checked the original paper of monotonic groves of neural additive models (MGNAMs). I want to know if $f_{1, 2, 3}, f_{1,2}, f_{4}, ..., f_{10}$ are all neural networks. If so, I am a little curious whether the $f_{4}, ..., f_{10}$ here are all single-dimensional input neural models?

See more questions and comments in Weaknesses.

---

### Official Review · Reviewer_HG66 · 2023-10-31

**Soundness:** 3 good
**Presentation:** 2 fair
**Contribution:** 2 fair
**Rating:** 3
**Confidence:** 4

**Summary:**

This manuscript presents two new values, the Monotonic Shapley value (MShap) and a Generalized Monotonic Shapley value (GMShap).  These are designed to overcome what the authors regard as a failure of the 'baseline' Shapley value (BShap), namely that it does not satisfy an axiom called 'average strong pairwise monotonicity' (ASPM).

Theoretical results show that MShap and GMShap satisfy novel variants of the usual symmetry and dummy axioms.  Some empirical results are presented as well: (local) Shapley and GMShap values are compared for an observation in a Kaggle credit score dataset; the "proportion of violations of strong pairwise monotonic features" in a COMPAS recividism and a Twitter bot dataset are compared across Shapley and GMShap.

**Strengths:**

**originality**

**quality**

As far as I can tell, the results presented are correct.

**clarity**

**significance**

Overall, I think that the project of interpreting complex ML models is an important one.  Thus, I think that efforts to improve our ability to do so are important.

**Weaknesses:**

For all its flaws, the Shapley value has two main strengths:
1. a clear derivation from easy-to-understand and generally compelling axioms;
1. a clear interpretation as an average change in a prediction caused by knowledge of a feature's specific value.
This are both important in general, but particularly in a measure that is supposed to aid understanding of complex models: an uninterpretable tool will be of limited use in interpreting models.

Thus, my main concern about this paper is that it slips on both fronts:
1. I do not have a clear sense of what the 'dummy(b)' and 'symmetry(b)' axioms are, other than that they are not quite the intuitive meanings.  The textbook way of introducing new terms is to show examples and counter-examples, rather than just definitions.  That is not done here.
1. as a result, I do not have a 'one line' explanation of what the MShap or GMShap measures.  Thus, I can see how they rank features, but do not know what that means.
Relatedly, I do not have a clear intuition for ASPM either.  Thus, I do not see why BShap's failure to satisfy it is of concern.

To the extent that I understand the pairwise monotonicity measures, they seem scale dependent.  Thus, in the examples given, if $x_{\beta}$ is measured in centimeters, and $x_{\gamma}$ in meters, then $x_{\beta} + c$ adds centimeters to a measurement, while $x_{\gamma} + c$ adds meters to it.  While this is an obvious example, and easily avoided, the whole point of attribution methods is to use them in situations for which the analyst does not have strong intuitions.  One obvious solution would be to just work on normalized variables.  In any case, this is an issue that the paper needs to address, and resolve.

If I were trying to establish ASPM as an important axiom to satisfy, I think I would want to show clear examples in which a result without ASPM was clearly 'wrong' in some very intuitive way.  This is not convincingly done in the paper: the opening motivation is merely an assertion that
> in credit scoring, the number of past dues more than two months should be more significant than the number of past dues between one and two months.
Is this true?  What do interpretable credit scoring models (of which there have been many since the FICO explainable AI challenge) say?  To me, it seems conceivable that "between one and two months" gives more contemporary insight into a creditor's ability to repay than "more than two months does".

Further, if I was then using a non-interpretable credit scoring model, and wanted to use a Shapley-style attribution layer to explain it, I would compare the sizes of the [Shapley] values themselves, rather than looking at whether they satisfied an ASPM axiom.

Later on, in $\S 4.1$, it seems that a somewhat different motivating concern is mentioned, namely that - when features are not independent - it may not make sense to consider the marginal change from $v(S)$ to $v (S \cup {i})$.  If this is the motivating concern, then the manuscript should lead with it, and introduce at least the highlights of the literature on this problem (e.g. the use of conditional v unconditional baseline values, for in/out-of manifold Shapley values, or the two papers on causal Shapley values at NeurIPS 2020, Heskes et al. and Frye et al.).

In terms of presentation, I found the manuscript casual ("by considering a variety of axioms", "capable of preserving many desired axioms", "we assume that $x_{\beta}$ has greater significance", "generalise the game with general features ... and more complex features can be generalized if necessary", "We treat $x_P$ as a single features since they usually describe the same feature"), leading me to wonder how tight a grasp of the material the authors themselves had.  Further, as a minor point, notation was inconsistent: in (5), we are told that $c > 0$, while, in (6), $c \in \mathbb{R}^+$.

Finally, in terms of the examples, it seemed to me that:
1. Kaggle credit scoring: without knowing how the features actually change default, I have no idea how feature attributions should be ranked.
1. Compas, Twitter bots: I have no idea why we're looking at number of violations of ASPM, rather than just comparing the Shapley values of the felony and misdemeanour features.

**Questions:**

1. The manuscript stands or falls, in my view, on whether or not ASPM is an important property of a feature attribution method.  Can the authors present a stand-alone argument about why satisfying ASPM is important?
1. Relatedly, how do the authors complete this Shapley-style sentence: "because the GMShap for this feature is twice as large as the GMShap for that feature, we can conclude that the first feature's impact on predictions is..."?